# An Online Distance Tracker for Verification of Robotic Systems’ Safety

**DOI:** 10.3390/s23062986

**Published:** 2023-03-09

**Authors:** Esra Guclu, Özlem Örnek, Metin Ozkan, Ahmet Yazici, Zekeriyya Demirci

**Affiliations:** Department of Computer Engineering, Eskisehir Osmangazi University, Eskişehir 26480, Turkey

**Keywords:** minimum distance calculation, safety verification, octree-based map, robotic systems

## Abstract

This paper presents an efficient method for minimum distance calculation between a robot and its environment and the implementation framework as a tool for the verification of robotic systems’ safety. Collision is the most fundamental safety problem in robotic systems. Therefore, robotic system software must be verified to ensure that there are no risks of collision during development and implementation. The online distance tracker (ODT) is intended to provide minimum distances between the robots and their environments for verification of system software to inspect whether it causes a collision risk. The proposed method employs the representations of the robot and its environment with cylinders and an occupancy map. Furthermore, the bounding box approach improves the performance of the minimum distance calculation regarding computational cost. Finally, the method is applied to a realistically simulated twin of the ROKOS, which is an automated robotic inspection cell for quality control of automotive body-in-white and is actively used in the bus manufacturing industry. The simulation results demonstrate the feasibility and effectiveness of the proposed method.

## 1. Introduction

Robotic systems are widely used in factories with the latest industrial development. They accomplish tasks individually or work together with people within dynamic environments. However, the usability of robotic systems in the industry highly depends on proving that these systems are safe and reliable. For this reason, robotic systems are expected to comply with the standards [1,2,3] established in this field. To ensure reliability, in addition to fulfilling these standards in robotic systems, methods and tools for verification and validation (V and V) should be developed. The safety standards [1,2] for industrial robots list the methods and specific performance requirements defined as necessary for the safety of the robotic system as part of the verification and validation of safety requirements and protective measures. Most of the requirements are related to distance. Distance is an essential element for robotic applications and is widely used for various purposes such as collision detection, collision avoidance, online motion planning, etc. Thus, the minimum distance calculation is a necessity for providing a safe working environment for both humans and robots and for the robot to perform its functions correctly. Moreover, the minimum distances are assets for the verification of the robotic system in terms of reliability and compliance with the safety standards. The expression “The robot can maintain a determined speed and separation distance (SSM)” provided under collaborative operation requirements [1] is related to distance.

Robots are used for production or transportation in the industrial environment. A robotic system must not be subject to any collision between robots and objects in the environment. So in studies with distance calculation, the general aim is usually to prevent collisions for safety at the design or verification phases of the system. We aim to develop a method for online distance tracking to verify the safety of robotic systems. The main contributions of this study are as follows: (i) A framework utilized for the purpose of online distance tracking between the robot and its environment from both/either the 3D mesh files of the entities and/or depth sensor readings. (ii) The representation of robots and their environment respectively with cylinders and an octree-based map, and the minimum distance calculation over representations of robots and their environments with a reduction in processing time by the proposed technique in which the voxels of the map in the neighborhood of the robot are only searched and used to calculate distances.

The rest of the paper is organized as follows: Section 2 provides related works. Section 3 describes the proposed method of an online distance tracker. Section 4 presents the application and results. Finally, the conclusion is provided in Section 5.

## 2. Related Work

This work is related to online minimum distance tracking between robots and their environments for the safety verification of robotic systems. The proximity between the robot and its environment is determined by the minimum distance computation between them. Generally, the objects in the environment are represented by convex geometries for fast distance calculations. In the robotics literature, Gilbert–Johnson–Keerthi (GJK) [4] and Lin–Canny (LC) [5] are the two main algorithms employed by researchers to calculate minimum distance. GJK is more robust and it uses less memory than LC [6,7]. Therefore, compared with LC, GJK is widely used in computing distance for convex objects. In a recently proposed approach [8], human body joints are obtained by using a skeletal tracking algorithm via an RGB-D sensor. Then, both the human body and robot arm are represented by capsules and the minimum distance is calculated using the GJK algorithm. Another method proposed by Han et al. [7] also uses GJK to estimate the minimum distance between a robot manipulator and an obstacle. Firstly, point cloud data of the dynamic obstacle is acquired from Kinect-V2 and converted to an octree. Secondly, a convex hull is created using the octree and finally, the closest distance is calculated between the convex hull and the cylinder representing the robot arm. Unlike our study, the obstacles can be represented by a convex hull. However, it only considers the human arm as an obstacle and shows performance results for it. It does not evaluate the problems of representation by a convex hull that may arise in a structure that the robot reaches into, such as a bus skeleton, as in our study. Stueben et al. [9] computed the closest distance for collision avoidance. In the approach, a robot and an obstacle are modeled as geometric primitives. Then, the GJK algorithm is applied to find the nearest points between the robot and the obstacle surfaces. The robot stops if the closest distance exceeds a safety threshold. This paper just focuses on the control of robots considering the minimum distance between humans and robots. Therefore, there is a lack of description of a technique for representations and distance calculation. A real-time collision detection method presented by Coulombe and Lin [10] employs an extended version of the GJK algorithm. The robot arm is represented with convex hulls and the extended GJK is used to estimate the minimum distance between the convex hulls. In this study, the mesh models of the robots taken from STL files are used for collision detection, not minimum distance calculation. The mesh model is considered a convex hull and no preprocessing was required to obtain a convex hull.

Besides GJK, some different algorithms have been proposed for distance calculation. Mronga et al. [11] extended the Kinematic Continuous Collision Detection Library (KCCD) [12], which was developed to detect collisions between rigid bodies in real time, for computing the minimum distance between a robot arm and an unknown obstacle. Point cloud data gathered from an RGB-D sensor is converted into convex hulls and provided to KCCD as input. Then, distance computation between convex hulls is performed with extended KCCD. The demonstration of the method is implemented by the convex hull of the point cloud belonging to the human arm. It is not suitable for large obstacles such as a bus skeleton because the convex hull of the point clouds may encapsulate some obstacle-free spaces as a part of the obstacle. An analytic approach is presented by Safeea and Neto [13] for calculating the minimum distance between a human and a robot arm. A laser scanner and five inertial measurement units (IMUs) are used to correct the measurements for human representation. The human upper body is modeled with IMUs. The robot arm and human lower body are represented using the laser scanner. Moreover, the entire human body and the robot arm are covered by capsules. The minimum distance between capsules is calculated by QR factorization. This method is specifically developed for the calculation of the minimum distance between humans and robots, not for an obstacle in the environment.

Flexible Collision Library (FCL) [14] is a GJK-based framework. It allows to perform collision and distance queries for various object shapes (box, cylinder, mesh, point cloud, etc.). Moreover, it provides the Robot Operating System (ROS) [15] interface to employ different types of robots. It is popularly used in collision avoidance and path planning methods. One of the approaches based on FCL is proposed by Wittmann et al. [16]. They developed a motion planning framework that uses FCL to perform distance computation and collision checking. A robot arm and a dynamic obstacle are represented by a capsule and a sphere, respectively. During the motion planning, the minimum distance is computed between the capsule and sphere using FCL. Another motion planning study presented by Grushko et al. [17] aims to improve human–robot collaboration by employing FCL to compute distance queries. At first, an OctoMap [18] is created via three RGB-D sensors. Then the right and left hands of humans are represented as spheres. The nearest points are calculated between the OctoMap and the spheres, and the vibration is sent to the closest hand. This study focuses more on hand tracking. It is not suitable for use in calculating the distance between the robot and the obstacles in the environment. Hu et al. [19] proposed a real-time motion planner tested in a mobile robot. The robot’s body is modeled as boxes, also three cylinders are used per arm. The obstacle can be represented as a box or an octree preferably. FCL is used for collision checking. In this study, the distance calculation is used for online motion planning. However, a single small-sized obstacle was used in the tests and no performance evaluation or solution was presented for the large-sized obstacles. Di Castro et al. [20] developed a collision-avoidance system that performs distance computation between robot arms or between a robot arm and an obstacle. The robot arms are modeled as boxes. Point cloud data of the obstacle is obtained from an RGB-D sensor. The point cloud can be used directly or converted into an octree. Minimum distance computation is performed via FCL. This paper deals only with the distances between robots and humans. However, the performance for minimum distance calculation between the robot and the large structures is not evaluated.

Most of the methods from previous studies are based on utilizing representations by primitive convex geometries and convex polygons. However, the convex hull of large structures such as bus skeletons similar to the ones provided in Section 4 represent the outer surface of the structure. When the robot enters into the structure from the holes, the distance v-calculation fails. To solve this problem, we need to calculate convex hull decomposition which fills the interior space of the structure [21]. However, the computations for the convex hull decomposition take a long time [22]. The proposed method uses the representation of the environment in an octree-based map. This map can be constructed by the provided CAD models and can be updated by the point clouds which are captured periodically during runtime. The minimum distance calculation is employed directly with the map without making preprocess on the map. The calculation load caused by the large number of voxels in the map is resolved by the bounding box approach. 

Valori et al. [23] discussed a cross-domain approach toward the definition of step-by-step validation procedures for collaborative robotic applications. The procedures are defined and connected to safety regulations and standards. The procedures are almost manual. Similarly, Fischer et al. [24] emphasized that the simple safety property that the robot shall not collide with any obstacle is an obvious property to verify. Our proposed method supports the automatization of robot verification by providing a framework for minimum distance tracking where distance is the main parameter for most of the safety requirements in standards. Ferrando et al. [25] introduced ROSMonitoring, a framework to support Runtime Verification (RV) of robotic applications developed using the Robot Operating System (ROS). The framework monitors the state of the software of the robotic system under test and implements the verification of whether the system fulfills the requirements. The verification tools for robotic systems such as ROSMonitoring need frameworks providing additional information about the system such as minimum distances. We present a framework utilized for the online distance tracking between the robot and its environment from both/either the 3D mesh files of the entities and/or depth sensor readings for the purpose of supporting the safety verification of robotic systems. 

## 3. Proposed Method

In the context of the work, the aim is to develop a method for online tracking of the distances between the robot and its environment. The method is intended to be used primarily for runtime verification of the robotic systems and secondarily to provide feedback to online robot motion planning. The architecture of the method is provided in Figure 1.

There are five artifacts which are composed of four inputs and one output. Two input artifacts are needed to create the configuration of the system from previously known information about robots and the environment, respectively, as parameter files of robots and CAD models of entities in the environment. The other two artifacts must be provided continuously within each period in which the distance calculation is employed. However, the provision of depth sensor data is selective depending on whether the environment is dynamic or static. In factories, generally, the workspace of the robotic systems is structured and surrounded by safety fences. In this case, the CAD models of the entities in the workspace in addition to the periodically updated robot joint values are sufficient for online distance calculations.

### 3.1. Map Construction

The representation of the environment is needed for distance calculation. The symbolic representation of the environment can be implemented by various types of maps. However, maps are usually used in robotics applications for different purposes. These can be provided as navigation, mission planning, path planning, exploration, or distance calculation. Moreover, in this study, a map has an important place for minimum distance calculation between the robot and its environment. Thus, map construction is one of the fundamental steps in the proposed method.

In map construction, a volumetric map is employed to represent the workspace of the robot. An octree-based occupancy map might be a good option for the representation of the environment. An octree is a tree data structure in which each internal node has exactly eight children [26]. Octrees are most often used to partition a three-dimensional space by recursively subdividing it into eight octants. The occupied octants are determined by the sensor readings or provided CAD models. Many software libraries are implementing an octree-based occupancy map; however, in this study, we used OctoMap [18], which is widely used in robot applications.

The map is represented by M. The map construction can be performed in two processes: Mapping by CAD models and mapping with depth sensing. Any process can be implemented and both processes can be employed. 

#### 3.1.1. Mapping by CAD Models

Mapping by CAD models is implemented by sampling points on the mesh of models. Thus, a point cloud, $P_{i}^{c}$, i∈ℤ ve $i\in \left[1,{𝓃}_{m}\right]$, is obtained per each CAD model indexed with i. Let ${𝓃}_{m}$ be the total number of CAD models stored as separate files. CloudCompare, which is an open-source project for 3D point cloud and mesh processing software, is a software tool with the capability of converting 3D mesh models into point clouds [27]. The number of points representing the model is determined by either specifying the total (approximate) number of points to be sampled (${𝓃}_{T}$) or the surface density (number of points per square unit) $\left({𝓃}_{U}\right)$, and the two parameters are reciprocal. CloudCompare is employed to convert ${𝓃}_{m}\;$ number of CAD models into ${𝓃}_{m}$ point cloud files. After obtaining point clouds for each model, the next step is to insert point clouds into the 3D occupancy map. OctoMap framework provides functions to integrate the point clouds into a volumetric occupancy map. Thus, the known objects in the workspace are represented in the occupancy map. If the workspace of the robotic system is surrounded by fences and statically structured, then the constructed map would be sufficient for reliable distance calculations. Since the use case demonstrated in this study is an industrial robotic inspection system that has a statically structured environment and fences surrounding it, the implementation of the methods just involves mapping by CAD models. Figure 2 includes a demonstration of map construction from CAD models. In Figure 3, one of the parts from the bus is selected, and respectively shows the mesh model, rendered view, and point cloud of the part. Then, the demonstration of a selected part on the occupancy map of the bus is provided in Figure 4.

#### 3.1.2. Mapping with Depth Sensing

The other process for map construction, which is represented as an occupancy map updater in Figure 1, involves point clouds taken by the depth sensors. This process is straightforward. The point clouds taken from the depth sensors need to be transformed into the world coordinate system before adding to the map. If the sensor coordinate frame is also provided in addition to the point cloud, the free voxels are obtained by clearing the space on a ray from the sensor origin to each endpoint in the set of point clouds. Thus, mapping from the depth sensor readings represents free spaces in addition to the occupied spaces, unlike mapping by CAD models which just determines the occupied spaces. It is better to use this process for dynamically changing environments as well as the cases in which CAD models are unavailable. Moreover, it is possible to use both processes such that a map is firstly created from CAD models, and then the map is updated by depth sensor readings.

### 3.2. Robot Representation

The representation of the robots with convex primitive geometries dramatically reduces the time for distance calculation. These geometries may be capsules, spheres, or cylinders that are used in this study. The cylinder is represented by two parameters which are the height $h=\left|p_{1}p_{2}\right|$ and radius r as seen in Figure 5. 

The cylinder axis c∈ℝ3 is oriented along the z-axis of the cylinder frame Oci. Each cylinder representing any part of the robot needs to be defined with respect to a reference frame, Ow, which is also the reference frame for other objects in the environment. The position and orientation of the cylinder can be represented by Tciw. However, the cylinders are fixed to robot link(s) in order to cover them while they are moving. For distance calculation to be effective and efficient, the quantity and dimensions of the cylinders must be properly determined. Therefore, the number and the size of cylinders should be chosen to cover as many parts and volumes as the physical structure of the robot. The number of cylinders and their positions may differ in various industrial setups. If each cylinder representing links of the robot is fixed to any link frame of the robot, say Oqi, and the fixed transformation Tciqi is defined, then the position and orientation of the cylinders can be defined with respect to Ow by the knowledge TqiwQ for any robot joint configuration. When Q=q1⋯qnT is any configuration of the robot, qi,i=1,⋯,n, are the joint values, and n is the number of joints for the robot. 

For each type of robot, the proposed method needs information about the cylinders provided by
(1)$$C_{i}=\left\{T_{C_{i}}^{q_{j}},h_{i},r_{i}\right\},\;i\in \left[0,m-1\right],\;\mathrm{and}\;j\in \left[0,n-1\right]$$
where $T_{C_{i}}^{q_{j}}$ is the homogeneous transformation from ith cylinder to jth joint frame, $h_{i}$ is the height of ith cylinder, and $r_{i}$ is the radius of ith cylinder. The robot, which has n joints, is represented by m cylinders. In order to calculate the distances, the parameters in (1) must be defined once for each type of robot used in a robotic system. 

#### 3.2.1. Primitive Geometry Extraction

In the implementation of the method, we propose to use Extensible Markup Language (XML) to describe the parameters for the representation of any type of robot. Thus, the distance calculation software is isolated from the dependency on a particular robot. The software works for any type of robot whose parameters are provided in XML format. An XML may contain:Robot brand, model, and root frame name;Each cylinder’s id, name, half of the height in meters, and radius in meters;For each cylinder, the associated robot frame’s id, name, translation, and rotation from the cylinder frame to the robot frame.

The sample XML for the OTOKAR’s robot which is ROKOS is provided in Figure 6.

#### 3.2.2. Primitive Geometry Position Update

Each item in the environment must have the same reference frame which includes the position and orientation of the item. At this point, the reference frames belonging to the links of the robot arm must be transformed into the world coordinate system. Figure 7 shows reference frames of the robot links and their cylinder representations.

### 3.3. Minimum Distance Calculation

The minimum distance calculation is performed between the representations of robot links and the obstacles in the environment. Figure 8 shows the block diagram of the minimum distance calculation. The robot links are represented by cylinders, Ci, and an octree-based 3D occupancy map, M, represents the workspace of the robot. M is composed of a set of free (Vf) and a set of occupied (Vo) voxels. A computational cost-efficient minimum distance calculation is implemented by decreasing the number of voxels in Vo, which is involved in the minimum distance calculation. Therefore, a minimum bounding box, BBmin=βmin∈ℝ3,βmax∈ℝ3, is defined to enclose the cylinders representing robot links. βmin and βmax are the minimum and maximum corners of the rectangular prism, respectively. Then, the bounding box is extended by a scalar δ ϵ ℝ, such as BBmine=βmine∈ℝ3,βmaxe∈ℝ3, where βmine = βmin−δ.1 1 1T and βmaxe = βmax+δ.1 1 1T. Thus, the minimum distances for each cylinder can be searched over the occupied voxels in BBmine instead of searching all the occupied voxels in M. Let Vobb be a set of occupied voxels such that Vobb⊃Vo and every center of voxels in Vobb are in BBmine.

FCL [14] is used to calculate the minimum distance dmini∈ℝ between the cylinder Ci and voxels in Vobb. Let Dmin = dmini ∀i ϵ 1, m where m is the number of cylinders. In this study, octree voxels are represented as boxes, and robot arms as cylinders. However, defining the collision object for all voxels in the octree and calculating the distances between cylinders and voxels is not cost-efficient. In this case, the second phase of the proposed method comes into play. A bounding box is created around the robot arms (see Figure 9). The dimensions of the bounding box can be determined by considering the distance needed for verification or reaction of the robotic system. Then, the occupied voxels of the octree map only in the bounding box are defined as collision objects which are subject to the minimum distance calculation against robot links. This approach allows for making a more cost-efficient calculation. Thus, distance calculations between the cylinders and the voxels of the map, which represent the model of the environment, are performed using FCL. The pseudo-code of the method used in minimum distance calculation is provided in Algorithm 1.
**Algorithm 1.** Minimum distance calculation**Descriptions:**𝓃v : number of occupied voxels Vo in MM: 3D occupancy map. Voi⊃ M, i∈1,𝓃v m: number of cylinders representing the robotCi: cylinders representing the robot, i∈0,m−1βmin: minimum corner position of the bounding boxβmax: maximum corner position of the bounding boxδ: a scalar number for extending the bounding boxDmin: a set including minimum distances between cylinders and occupied voxels in the map, Dmin = dmini ∀i ϵ 1, mVobb: a set of occupied voxels in BBmineδ=mDistθ1,θ2: a function that takes two geometric objects, say θ1 and θ2, and returns the minimum distance, say δ∈ℝ, between θ1 and θ2**Inputs:**M, Ci, βmin, βmaxδ**Outputs:**Dminβmine←βmin−δ.111Tβmaxe←βmax+δ.111T Vobb←∅**PART A:****for**∀i∈1, 𝓃v**do**   **if**Voi in BBmine **then**      add Voi to Vobb   **end if****end for****PART B:****for**
∀i∈0, m−1 **do**    dmini←mDistCi,Voabb, for any Voabb∈Vobb     **for**
∀Vobbb∈Vobb∖Voabb **do**       dtempi←minDistanceCi,Vobbb       **if**
dtempi<dmini **then**          dmini←dtempi       **end if**    **end for**    add dmini to Dmin**end for**

## 4. Application

### 4.1. System Description

The proposed method is implemented for the use case with the name of an automated robot inspection cell for quality control of automotive body-in-white (ROKOS) in the VALU3S project [28]. The robotic inspection cell carries out quality control by utilizing the camera system mounted on the tips of two cartesian robots on both sides of the vehicle body. The data obtained from the CAD data of the large-bodied vehicle are compared with the actual data obtained from the camera system, and the presence/absence check for the parts composing the vehicle is performed. The ROKOS is produced for the quality control of a bus chassis which includes 2909 body parts. The ROKOS contains two cartesian robot arms and a bus skeleton. The robot arms follow a trajectory and take pictures of the skeleton at predefined positions. 

The application of the proposed method is achieved by a simulation-based robot verification testing tool (SRVT) [29]. The SRVT is a good representative of the ROKOS. It stimulates the ROKOS from task planning to 3D simulation with a dynamic engine and 3D visualization with the Gazebo Simulation environment [30]. In the proposed method, just one cartesian robot arm is used instead of two arms of the existing system. During the movement, the online distance tracker calculates the minimum distance between the links of the robot and the bus skeleton. Figure 10 shows the real (a) [31] and simulation environments (b–d).

### 4.2. Implementation

The ODT is implemented in ROS Noetic with C++. In addition, FCL 0.7.0 is used for distance calculation, and the OctoMap library is employed for 3D octree-based occupancy mapping. The system is tested on an HP-Z840-Workstation with a 32-core Intel^®^ Xeon(R) CPU at 2.40 GHz running Ubuntu 20.04.2 LTS.

#### 4.2.1. Map Construction by CAD Models

The map is constructed with CAD models of the parts of the bus. The body of the bus is composed of 2909 parts. In addition, two parts are used for support under the body of the bus. Therefore, a total ${𝓃}_{m}=2911$ CAD models in STL format are included in the occupancy map.

Figure 11 shows the flow diagrams for the mapping by CAD models. The CAD files are taken in STL format. The meshes of each CAD file are sampled points and the number of ${𝓃}_{m}$ point cloud files are generated. The sampling of points is implemented by the surface density method with ${𝓃}_{U}=20$ points per square unit. For the map, an octree structure is used with a 0.05 m resolution. Figure 12 shows the map of the bus.

#### 4.2.2. Representation of ROKOS

Six cylinders, $C_{i},\forall i\in \left[0,5\right]$, are defined for the representation of the links of the robot as seen in Figure 13. Moreover, the parameters of the cylinders are provided in Table 1.

#### 4.2.3. Minimum Distance Calculation Results

Minimum distance calculation is the most critical stage for the computational cost of the method since the calculations are performed simultaneously during the system operation. Therefore, some analyses are provided before the demonstration of the implementation of the method. 

First, we examined the reasons for the high computational costs of the minimum distance calculation. Then, two tests were conducted, leading to the computationally cost-efficient method. The system provided in Figure 10 was used for the tests. 

In the first test, PART A of Algorithm 1 is not implemented.$\;V_{o}^{bb}$ is entirely composed of all $V_{o_{i}}⊃\;M,\;i\in \left[1,{𝓃}_{v}\right]$. In the second test, PART A of Algorithm 1 is implemented with a bounding box $BB_{min}^{e}$ where $\beta_{min}={\left[\begin{array}{ccc} 2.8 & -2.0 & 0.4 \end{array}\right]}^{T}$, $\;\beta_{max}={\left[\begin{array}{ccc} 4.2 & 2.1 & 4.8 \end{array}\right]}^{T}$, and δ=0. The minimum and maximum points of the bounding box ($\beta_{min}$ and $\beta_{max}$) are determined by considering the farthest point that the robot arm can reach in the environment. Figure 14 shows the test environment including an occupancy map, a cylinder, and a bounding box. Test results are provided in Table 2. If the bounding box is used, the number of voxels involved in distance calculation decreases, and consequently the total time for the calculation of the minimum distance between a cylinder and the complex objects.

While the ROKOS is performing the provided inspection task, at the same time, the ODT is also active. 

Figure 15 shows the minimum distances between cylinders and the map in a given period. During the period, the closest distance of *C_0_* does not change noticeably as it only shifts on the *x*-axis parallel to the bus. *C_1_* moves on the *x*-axis and *z*-axis, and the minimum distance is over 1 m throughout its motion. The cylinders *C*_2_, *C*_3_, *C*_4_, and *C*_5_ move all three axes (*x*, *y*, *z*). The 110th s is the first time that all cylinders together obtain the closest distance to the bus. At this time, the nearest distances for *C*_2_, *C*_3_, *C*_4,_ and *C*_5_ are 0.34 m, 0.18 m, 0.13 m, and 0.31 m, respectively (see Figure 16c). 

The dimensions of cylinders are crucial for the representation of the robot links. The dimensions should be the smallest values that cover the links. Otherwise, the minimum distances cannot be reliable. For instance, at 210 s, the minimum distance of *C*_3_, which is −1, indicates a collision between *C*_3_ and the bus (see Figure 15). However, in the real system, there is no collision between the links of the robot and the bus. The reason is that the cylinder dimensions are chosen larger than they must be. The radius of the cylinder must be adequately close to the related link of the robot. Therefore, *C*_3_ is tested with three different radii to find the optimum radius. Figure 17 presents the minimum distances of *C*_3_ for three radii. As can be seen in the figure, the robot completes its movement without collision if a consistent radius is employed (r3 = 0.124). Even if the robot performs its movement without any collision with the r2 radius, it needs to be chosen as the nearest one for realistic distance results. Figure 18 shows the representation of the *C*_3_ with the new and old radii.

## 5. Conclusions

This paper presents an online distance tracker (ODT) to verify robotic systems’ safety. It uses FCL to calculate minimum distances between a robot and a bus skeleton which are represented as cylinders and occupancy maps, respectively. Unlike the other approaches which use the convex hull representation of objects for the distance calculation, the representation by occupancy map enables the minimum distance calculation even if the robot moves into the object. The ROKOS, which includes a non-convex object in its environment and is used to demonstrate the proposed method, is the best fit to show the advantage of the method. However, the occupancy map consists of too many cubic geometries, which are voxels, causing computational costs. To decrease the computational costs, a bounding box is defined around the robot, and the minimum distance is searched just for the voxels in the bounding box. In this manner, the minimum distance computation is performed only between cylinders and these voxels inside the bounding box. According to experimental results, it can be seen that the processing time of the minimum distance calculation is significantly reduced by the bounding box approach. ODT is implemented in a simulation environment. It is intended to use the safety verification of robotic systems. During the robot’s motion, it provides the minimum distances to keep the system informed. Using these distances, robotic systems can be verified for a safe working environment. Moreover, in a newly developed robotic system, verification for the reliability of the motion planning algorithms used/developed can be performed automatically.

ODT is implemented and tested in a robotic system that has a static environment. For future work, the occupancy map of the environment can be updated in real time using depth sensors located around the workspace. Thus, ODT can work with systems that have dynamic environments.

## Figures and Tables

**Figure 1 sensors-23-02986-f001:**
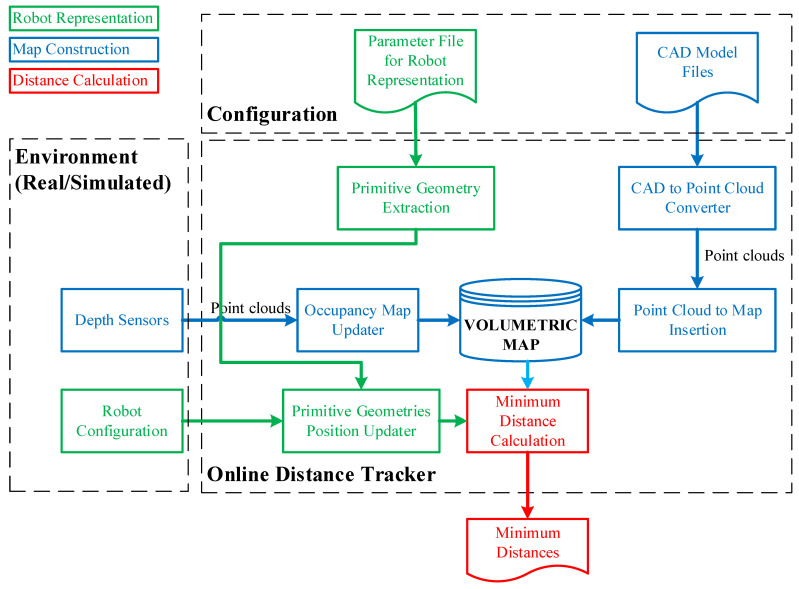
The architecture of the online distance tracker.

**Figure 2 sensors-23-02986-f002:**
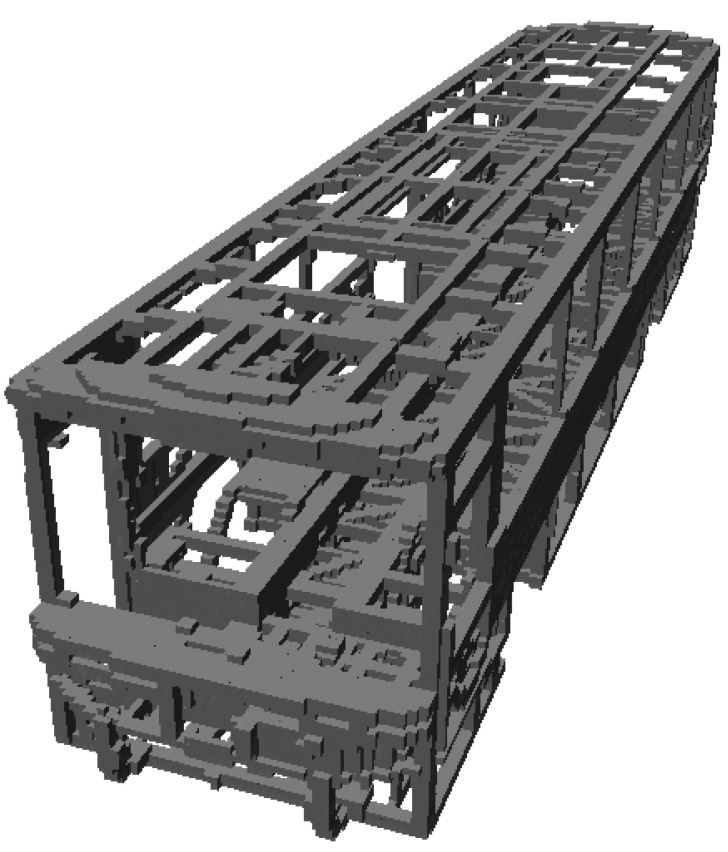
Demonstration of occupancy map constructed from CAD models.

**Figure 3 sensors-23-02986-f003:**
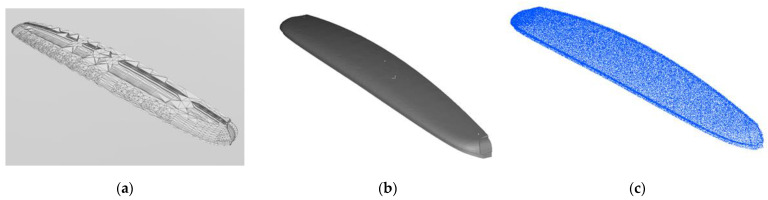
A part of the bus: (**a**) mesh model, (**b**) rendered view, (**c**) point cloud of the part from the bus.

**Figure 4 sensors-23-02986-f004:**
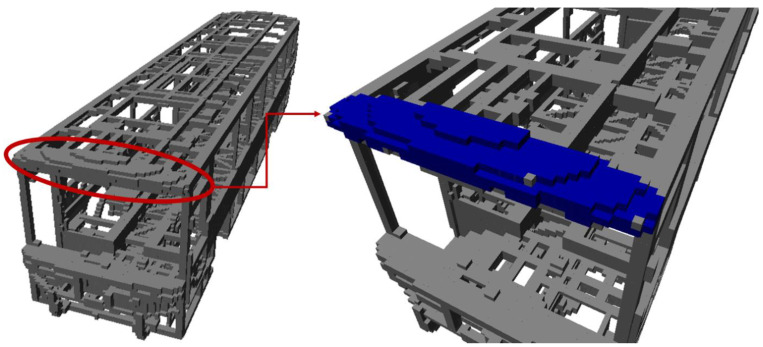
Demonstration of a selected part on the occupancy map of the bus (the selected part is indicated in blue on the occupancy map).

**Figure 5 sensors-23-02986-f005:**
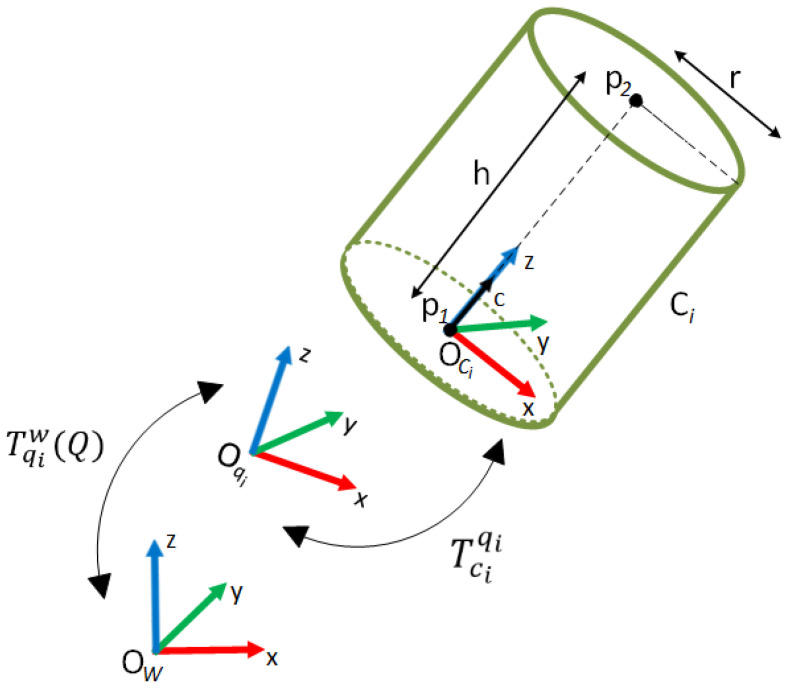
The geometry of the cylinder and its parameters.

**Figure 6 sensors-23-02986-f006:**
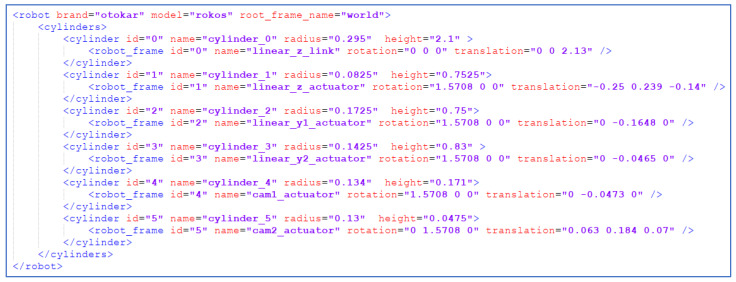
A sample XML for ROKOS.

**Figure 7 sensors-23-02986-f007:**
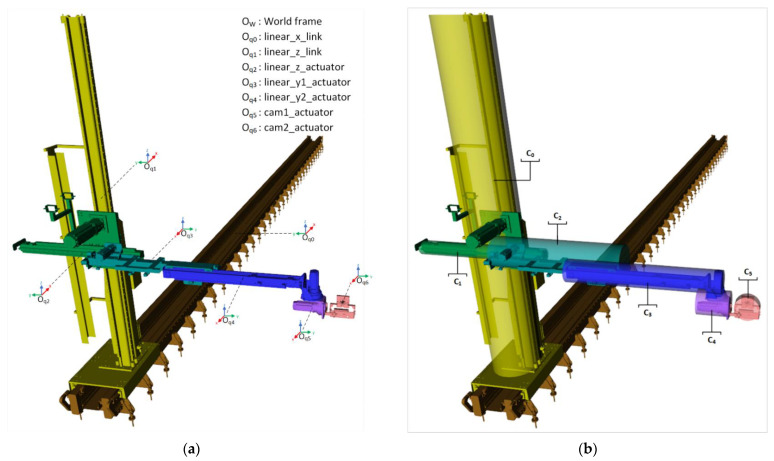
The robot frames and cylinder representations for ROKOS: (**a**) robot frames, (**b**) cylinders.

**Figure 8 sensors-23-02986-f008:**
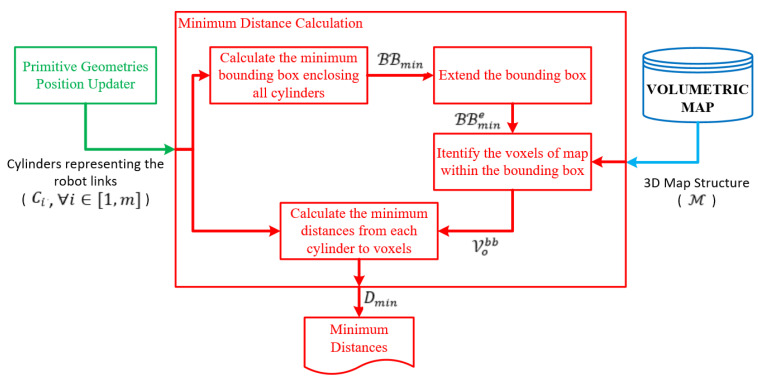
Minimum distance calculation.

**Figure 9 sensors-23-02986-f009:**
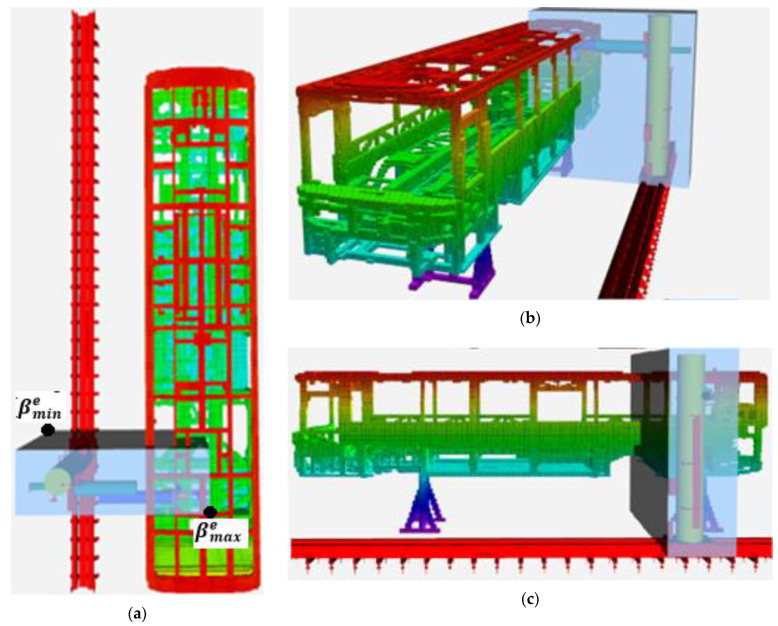
Demonstration of cylinder representation of the robot, volumetric map of the environment, and the minimum bounding box from different perspectives: (**a**) Top view, (**b**) cross view, (**c**) side view.

**Figure 10 sensors-23-02986-f010:**
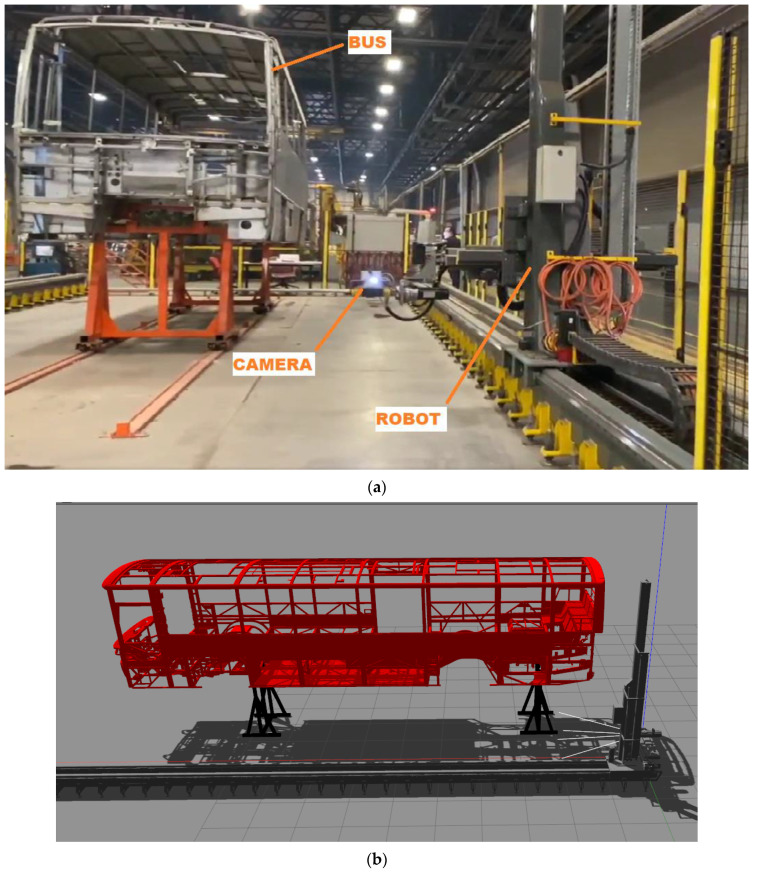
System description: (**a**) a photo of the ROKOS, (**b**–**d**) demonstration of the simulated environment with SRVT.

**Figure 11 sensors-23-02986-f011:**
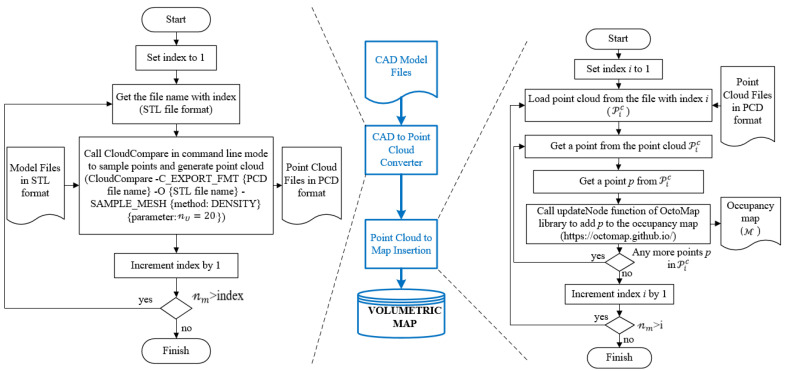
Flow diagram for mapping by CAD models.

**Figure 12 sensors-23-02986-f012:**
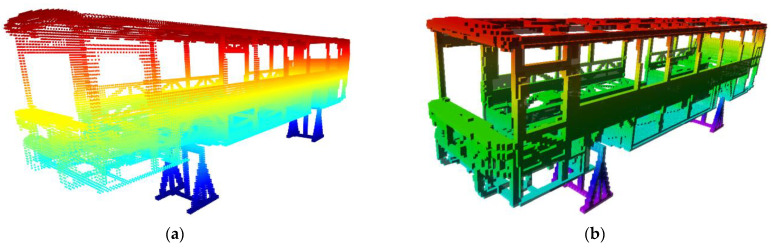
Mapping by CAD models for bus body: (**a**) point clouds, (**b**) 3D octree-based occupancy map.

**Figure 13 sensors-23-02986-f013:**
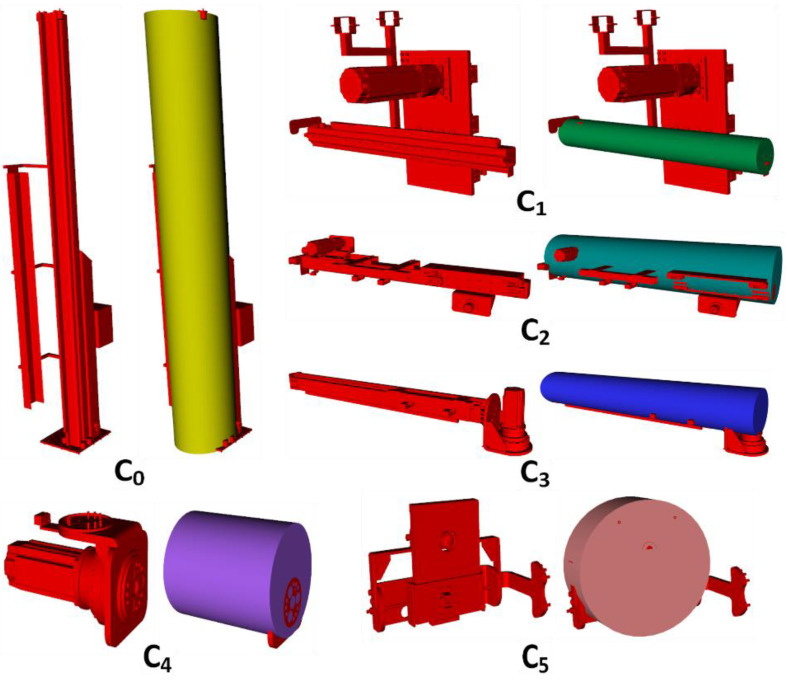
Links of the robot and its related cylinder representations.

**Figure 14 sensors-23-02986-f014:**
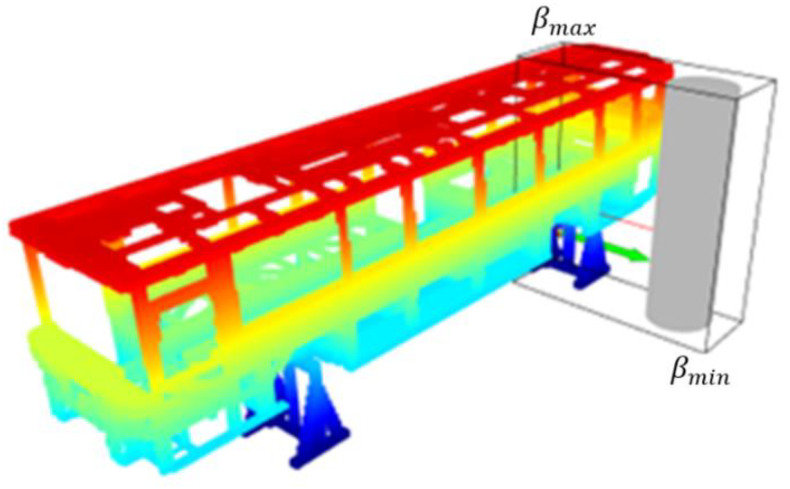
The representation of the test environment including the occupancy map, cylinder, and bounding box.

**Figure 15 sensors-23-02986-f015:**
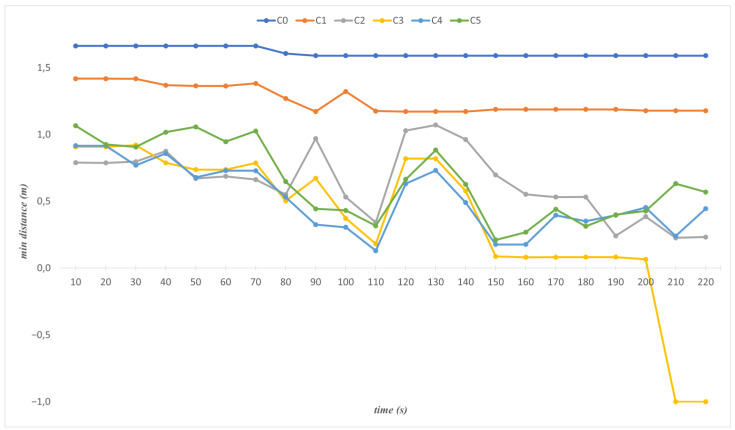
Minimum distances for each cylinder (*C*_0_ to *C*_5_).

**Figure 16 sensors-23-02986-f016:**
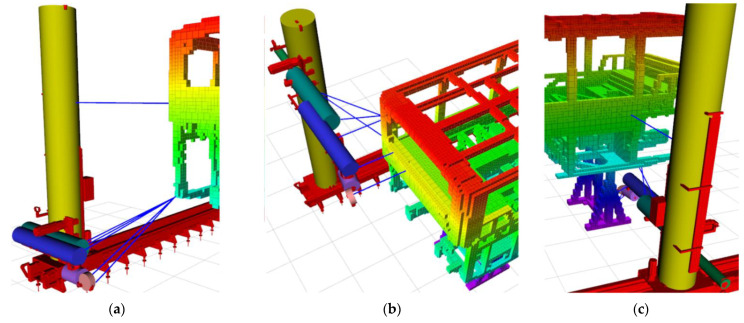
Visualization of the minimum distances of cylinders with blue lines at (**a**) 0 s, (**b**) 60 s, and (**c**) 110 s.

**Figure 17 sensors-23-02986-f017:**
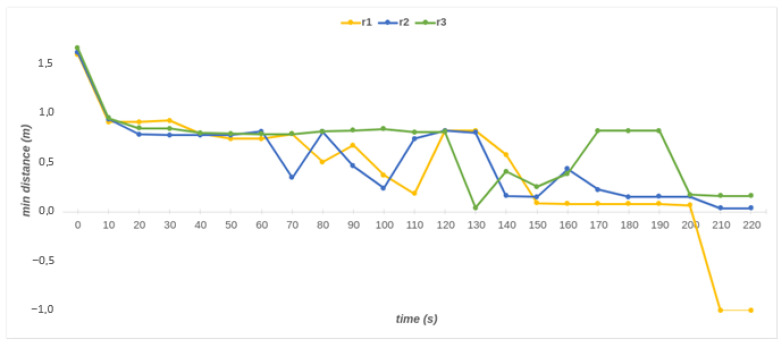
Minimum distances of *C*_3_ for three different radii. r1 = 0.1425, r2 = 0.135, and r3 = 0.124.

**Figure 18 sensors-23-02986-f018:**
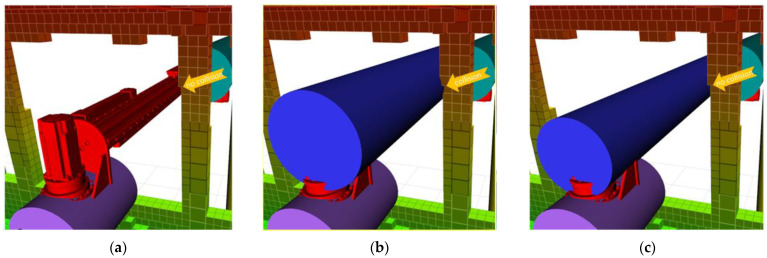
(**a**) Visualization of the 3rd robot link. (**b**) Representation of cylinder *C*_3_ with radius 0.1425. The radius causes a collision, unlike the real system. (**c**) Representation of cylinder *C*_3_ with radius 0.124. It provides the closest and most consistent motion to the real system.

**Table 1 sensors-23-02986-t001:** Parameters of the cylinders.

Cylinders	Robot Frames	Transformation Matrix	r (Meters)	h (Meters)
$C_{0}$	$O_{q_{1}}$	$T_{C_{0}}^{q_{1}}$	0.295	4.2
$C_{1}$	$O_{q_{2}}$	$T_{C_{1}}^{q_{2}}$	0.0825	1.505
$C_{2}$	$O_{q_{3}}$	$T_{C_{2}}^{q_{3}}$	0.1725	1.5
$C_{3}$	$O_{q_{4}}$	$T_{C_{3}}^{q_{4}}$	0.1425	1.66
$C_{4}$	$O_{q_{5}}$	$T_{C_{4}}^{q_{5}}$	0.134	0.342
$C_{5}$	$O_{q_{6}}$	$T_{C_{5}}^{q_{6}}$	0.13	0.095

**Table 2 sensors-23-02986-t002:** The results of tests for Algorithm 1 with and without bounding box.

	Time (s)
Distance calculation without $BB_{min}^{e}$	0.24238
Distance calculation with $BB_{min}^{e}$	0.017242

## Data Availability

Not applicable.

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
