# Peer review of "An Online Distance Tracker for Verification of Robotic Systems’ Safety"

_sensors, 2023, doi:10.3390/s23062986_

Round 1
Reviewer 1 Report
TThis paper describes an online distance tracker to verify the distances between a robot and the internal structure of a bus within an inspection cell. It uses cylinders representing robot links and bus occupancy maps in order to avoid collisions between them. Some specific comments on the article are as follows:
• Although it is understood what is in figure 6, I suggest that it be placed inside a box.
• It would be interesting to define the three views in Fig. 9 so that the reader can quickly interpret them.
• Check the content of the labels within figure 10(a). In this figure it is not clear where the camera is, as it should be assumed to be where it is indicated, but it is not properly perceived. Furthermore, it would appear that the camera is at the rear of the bus structure, and should be at the end effector of the Cartesian robot. Additionally, the Cartesian robot on the right is not clearly distinguishable, and the marking on its label seems to me to point elsewhere. Perhaps, better pictures should be taken and they should be properly labelled.
• In Fig. 11, one of the blocks in the flowchart on the left is empty, as are all the blocks in the flowchart on the right. This needs to be corrected.
• In Fig. 12, what is the meaning of the different colours in the bus structure?
• What is the criterion for selecting the minimum and maximum bounding box before running the simulations? Refer to the last paragraph on page 15.
• The unit of metres is missing in the distances in the text above figure 16, first paragraph, p.17.
• I think it should be explained a little better why with a smaller radius in the C3 cylinder there is no collision in relation to a larger radius for the minimum distances. Even if it is written that "It provides the closest and most consistent motion to the real system", which has been demonstrated, then with a larger radius, perhaps the motion of this link is further away from the bus. This needs to be explained better.
Author Response
We would like to thank the reviewer for carefully reviewing our manuscript, and making comments and suggestions that will contribute to the improvement of our manuscript.

Reviewer 2 Report
This study presents an approach to detect the minimum distance between a robot and its environment for collision detection purposes.
Remarks:
- The authors have enumerated the contributions of this paper at the end of the Introduction section. However, the contributions should be formulated related to the previous similar works and methods. What are the improvements of the proposed method related to the other similar methods presented in previous studies?
- For example, the authors claim that the convex polygons “are not feasible for the environments including such a structure that the robot needs to move inside”. Why the cylinder representation is better? This should be detailed.
- The entire paper mostly focuses on one application, the method does not seem general enough for a scientific paper. The method described Section 3 mostly focuses on a specific robot, using specific software tools ( e.g. Octomap, CloudCompare…)
- In the paper the authors deal with a Cartesian robot. Does the method is applicable to other types of robot, such as parallel arms or redundant robots?
- The authors claim in the title that they have developed an “online distance tracker”. However, in the Conclusions section the real time update of the point cloud is mentioned as future work. Moreover, “Occupancy map updater” and the “Primitive geometries position updater” are not detailed is section 3. The title and the contributions should be rephrased, as it seems that the developed approach is not (yet) online.
- Section 2: rephrase “[10] propose a …”
Author Response

(The authors gave the same response as above.)

Round 2
Reviewer 1 Report
I have checked the replies to my comments and they have been answered. I have no further comments to make.
Author Response
We thank the reviewers for their contribution to the improvement of our manuscript.

Reviewer 2 Report
The authors have addressed most of my remarks. However, in my view, the paper still remains an application paper rather than a scientific one. At the same time, it could contain good ideas for the readers that deal with similar practical applications.
One request still remains from the previous review: “the contributions should be formulated related to the previous similar works and methods. What are the improvements of the proposed method related to the other similar methods presented in previous studies?”
The quality of the presentation: The paper needs some more systematic presentation due to its length. Some important parts are hidden in short paragraphs in the middle of some subsections. E.g. in section 3 the elements of the diagram in Fig. 1 should be presented in separate subsections.
Author Response

(The authors gave the same response as above.)
